# Propofol-Induced Neurotoxicity in the Fetal Animal Brain and Developments in Modifying These Effects*—*An Updated Review of Propofol Fetal Exposure in Laboratory Animal Studies

**DOI:** 10.3390/brainsci6020011

**Published:** 2016-03-28

**Authors:** Ming Xiong, Li Zhang, Jing Li, Jean Eloy, Jiang Hong Ye, Alex Bekker

**Affiliations:** Department of Anesthesiology, New Jersey Medical School, Rutgers University, Newark, NJ 07107, USA; lizhangx1212@gmail.com (L.Z.); jingli_umdnj@yahoo.com (J.L.); eloyje@rutgers.edu (J.E.); ye@rutgers.edu (J.H.Y.); bekkeray@rutgers.edu (A.B.)

**Keywords:** propofol, neurotoxicity, *in utero*, apoptosis

## Abstract

In the past twenty years, evidence of neurotoxicity in the developing brain in animal studies from exposure to several general anesthetics has been accumulating. Propofol, a commonly used general anesthetic medication, administered during synaptogenesis, may trigger widespread apoptotic neurodegeneration in the developing brain and long-term neurobehavioral disturbances in both rodents and non-human primates. Despite the growing evidence of the potential neurotoxicity of different anesthetic agents in animal studies, there is no concrete evidence that humans may be similarly affected. However, given the growing evidence of the neurotoxic effects of anesthetics in laboratory studies, it is prudent to further investigate the mechanisms causing these effects and potential ways to mitigate them. Here, we review multiple studies that investigate the effects of *in utero* propofol exposure and the developmental agents that may modify these deleterious effects.

## 1. Introduction

Propofol is the most commonly used intravenous agent in current anesthesia practice. While it is generally considered safe in a mature brain, the safety of propofol exposure on a developing brain is questionable. Many general anesthetics such as isoflurane, sevoflurane, ketamine, and midazolam have been shown to cause adverse changes of the brain in neonatal animal studies. In recent years, evidence of the neurotoxic effects of propofol in the neonatal animals has also begun to accumulate. However, most current studies of propofol-induced neurotoxicity are derived from neonatal animal studies; much less is known about the effects of propofol exposure on a fetal brain. This review covers uncertainties about whether propofol would have a similar neurotoxicity profile in the fetal animal brain as in the neonatal animal brain. This review also addresses currently available methods to modify any potential neurotoxic effects of propofol on the fetal brain.

## 2. Current Understanding of Anesthetic Neurotoxicity

Since the 1980s, evidence of the potential neurotoxicity of anesthetic agents given to neonatal animals has emerged [1,2]. Numerous studies have confirmed that anesthetic agents given to neonatal rats lead to neurodegeneration and subsequent neurocognitive deficits and behavioral changes [3,4,5,6,7]. These results have been replicated in many different species, including primates [8,9,10].

Various factors contribute to the degree of neurodegeneration observed in these animal studies. Factors that influence the toxicity of anesthetic agents are characteristics of exposure, sensitivity of the brain and nervous system, and metabolic competency of the organism. Studies have demonstrated that evidence of neurodegeneration increases with longer duration, greater dosage, and higher frequency of anesthetic exposure [1,3,11]. A key factor of anesthetic-related neurotoxicity is the period or timing of the exposure in relationship to neuronal maturity. The vulnerability of neurons to exposure of anesthetics revolves around the concept of the brain growth spurt period (BGSP), a period of rapid formation of synapses (synaptogenesis) between the neurons of the central nervous system [12]. Significant evidence of neurodegeneration in rats occurs after exposure to anesthetic gases on postnatal day 7 (PND 7), which coincides with the peak of BGSP [3,6]. This observation is becoming a concerning issue with obstetrical and pediatric clinical anesthesia since the human BGSP is estimated to occur from the second trimester to three years of age [12,13]. Propofol is a relatively new general anesthetic that is being used in many of these procedures and is becoming a concern in the field of neurotoxicity.

## 3. Effects of Propofol on Neonatal Animal Brain

The findings from animal studies within the past ten years have shown that propofol exposure may induce similar neurotoxicity in neonatal animals of several species [14,15,16,17,18,19,20,21,22,23,24,25,26,27]. Evidence for neurotoxicity was supported by elevations in caspase-3 activity in neurons, an indication of neuronal apoptosis or death [17,18,19,20,21,22,23,24,25]. Caspase-3 is a protein that plays a central role in both the extrinsic and intrinsic apoptotic pathways. These neurodegenerative effects following propofol exposure have been observed in various animal models, including rhesus monkeys [28]. Furthermore, the observed neurodegeneration was associated with significant long-term behavioral deficits [18,26,27].

The role of excessive apoptosis has been a major research focus as a cause of the neurodegeneration that results from early propofol exposure [17,22,23,29]. Studies have found that anesthetic agents binding to the GABA and NMDA receptors during the BGSP may potentially trigger excess apoptosis and affect neuronal development [30,31,32,33]. A rationale for the increased vulnerability during this period is that exposure of anesthetics leads to excitation, rather than inhibition, of the GABA_A_ receptors during BGSP [34,35]. Other studies have investigated the specific secondary messengers that may be involved in the apoptotic cascade, such as the role of neurotrophins and the enzymes involved in its downstream cascade [17,22,23,29,36,37,38], the expression of tumor necrosis factor [17,23,29], and the activation of microglial cells in the brain. Other features such as suppression of neurogenesis [14,15] and alterations of dendritic spinogenesis may contribute to propofol-induced neurotoxicity [16]. More recently, study shows propofol promotes blood-brain barrier breakdown in the developing mouse brain [39]. The mechanism of neurotoxicity associated with neonatal propofol exposure has not yet been clearly identified. Nevertheless, these various mechanisms provide a positive indication that neonatal propofol exposure does cause neurotoxicity.

It is worth noting that the timing of propofol exposure of the young animal was very critical to detect neurotoxic effects [17,29], which has been verified in many studies involving other anesthetic agents [3,6]. The reason that anesthetic-induced neurotoxicity can arise in 7-day-old rats may directly relate to the contemporaneous peak of synaptogenesis. It is interesting to determine whether propofol exposure in younger rats (<7 days) can induce any detectable neurotoxicity. What stages of brain development are vulnerable to the neurotoxic effects of propofol? Would propofol exposure have an effect on the brain during gestation? Here, we review the few studies in the literature that address these questions.

## 4. Effects of Propofol in Fetal Animal Brain

Fetal exposure to general anesthetics may have a significantly different response to that of the neonatal brain. For example, Bai et al showed that neural stem cells were actually more resistant to neurotoxic damage than two-week-old differentiated neurons *in vivo* [40]. A number of animal studies showed that fetal exposure to volatile anesthetics does cause neurodegeneration and long-term change [41,42,43,44]. Anesthetic exposure of the dam also exposes the fetus to propofol since the highly lipid soluble propofol has been shown to transfer through the placenta [45,46]. However, whether this exposure leads to neurodegeneration remains largely unknown, but is an important research and clinical issue. 

Xiong and Li *et al.* have investigated the effects of fetal exposure *in utero* to propofol, as a model for surgery during pregnancy [36,47]. Xiong’s study assesses whether an intravenous infusion of propofol to a pregnant rat on gestational day 18 would lead to detectable neurotoxic effects in the brain of the offspring. Pregnant rats were exposed to propofol via tail vein infusion for 1 to 2 h on gestational day 18; they maintained normal pregnancy, and delivered offspring on day 21 to 23. Through this process, the study was able to control the dosage of propofol exposure. The initial study administered the subclinical dosage of propofol to pregnant rats, which was maintained at the infusion rate of 0.4–0.5 mg/kg/min for 0.5 to 2 h and produced a light sedation (defined as reduced activity but with intact righting reflex). The propofol-exposed pregnant rats were divided into two experimental groups to assess the early and long-term effect of propofol. Group one consisted of fetuses that were harvested via Cesarean section six hours after propofol exposure to investigate immediate effects of propofol exposure. Group two consisted of offspring of the pregnant rats that were allowed to undergo normal spontaneous vaginal delivery. These pups would then be followed for up to one month to look into long-term effects of propofol exposure.

The immediate effects of gestational propofol exposure were dependent on the duration of exposure. Gestational exposure to propofol for a short duration (0.5 h) at a light sedation (reduced activity but intact righting reflex; propofol infusion rate of 0.4–0.5 mg/kg/min) did not affect the levels of cleaved caspase-3 in the fetal brain. In contrast, 1- or 2-h propofol infusion at the same infusion rate (*i.e.*, providing 2× or 4× the dosage) significantly increased the activation of capase-3 in the fetal brain at 6 h post infusion (Figure 1). The increased apoptotic activity occurred in many different brain regions, with the heaviest concentrations in the cortex, thalamus, and hypothalamus regions. Double staining with anti-cleaved caspase-3 and anti-NeuN, a neuron-specific nuclear protein, demonstrated that the majority of the cleaved caspase-3 positive cells were neurons [36] (Figure 1). This widespread apoptosis in the fetal brains was documented via immunohistology and Western blotting. In addition, the fetal brains showed a significant activation of central neural inflammation including activation of microglial cells, which may serve as another possible mechanism for the observed neurodegeneration [36]. The results from Xiong *et al.* and Li *et al.* studies extended Creeley *et al.*’s primate studies in which the effects of propofol exposure in neonates and fetuses were compared. Creeley *et al.*’s study compared two experimental groups of rhesus monkeys: fetuses were exposed *in utero* on preconception day 120 and neonates were exposed on postnatal day 6. Different areas of neurodegeneration occurred in the two groups, with the damage more pronounced in the *in utero* exposed group [28]. Taken together, these studies are the first to demonstrate that prenatal exposure of propofol in animals may trigger similar or more pronounced neurodegeneration than that in neonatal animals [28,36,47].

The long-term effects of prenatal-propofol exposure are also very interesting. Besides the acute changes seen immediately following exposure to propofol, there seems to be an impact on the development of the offspring. Li *et al.* demonstrated developmental delays such as slower growth of body and brain, and slower maturation of neurological reflexes in rats that had been exposed to propofol prenatally (gestational day 20). However, these signs of developmental delays were no longer detectable on postnatal day 28 [37]. In addition to the apoptotic changes, the pups exposed *in utero* to light sedation with propofol on gestational day 18 also showed a significant decrease in synaptophysin on postnatal day 28 [47]. Synaptophysin expression is an indicator of ongoing synaptogenesis. The reduced synaptogenesis observed on postnatal day 28 may be one of the causes for long-term damages observed in pups exposed on gestational day 18 [47]. 

Addressing whether this persistent alteration of synaptogenesis would lead to behavioral and learning changes was an important question. Xiong *et al.* found that the propofol *in utero* exposed group possessed less exploratory activity and spatial learning deficits at postnatal day 28 in an eight arm radial maze than the control pups. These results suggested that the damage to the brain during fetal development can last longer than the transient activation of the apoptotic pathway [47]. This idea that the neurotoxicity induced by general anesthetics can lead to long-term behavioral change is not an isolated finding. Exposure of isoflurane and repetitive propofol administration to neonatal rats have been shown to cause long-term behavioral change [4,6]. However, given the complexity and different interpretations of animal behavioral studies, it is hard to conclude whether prenatal exposure of propofol can truly lead to long-term behavioral change at this time. More animal behavioral studies using different methods or different analytical strategies would help verify this conclusion. Nevertheless, Xiong *et al.* and Creeleys *et al.* research gives a clear indication that gestational propofol exposure may induce damage or at least alter the development of a fetal brain [28,36,47].

## 5. Reversal Agents: Their Use in Understanding the Neurotoxic Effects of Propofol Exposure in a Fetal Brain

Propofol-induced apoptotic activity in the fetal animal brain is still under active investigation. Finding the mechanisms of this neurotoxicity is a focus of future studies. The role of the fetal exposure of propofol on the significant apoptosis in the fetal brain from *in vivo* studies needs to be substantiated, especially upon recognizing other variables that may contribute to the apoptosis. They include: (A) propofol exposure via the placenta may cause hypo-perfusion or other physiological condition changes to both the fetus and mother; (B) The propofol solution used contained several other compounds such as EDTA, which could potentially be a neurotoxic compound; (C) The stress from these procedures, such as inserting an intravenous line, may also cause abnormal changes in the fetal brain. Those questions are only partially addressed in the previous experiments with the monitoring of the health status of the pregnant rats during the anesthesia and the use of control groups. A scientific model that may validate fetal exposure of propofol’s neurotoxicity but also limit the role of those previous variables is required for the analysis of possible ways to block or diminish the apoptosis under identical experimental conditions. Recently, many agents including lithium, erythropoietin, and dexmedetomidine (Dex) were shown to reverse the anesthetic-induced toxicity [48,49,50]. More recently, pre-administration of rutin, curcumin and Dex can attenuate neuronal apoptosis, showing significant protection against anesthetic-induced neurodegeneration and learning and memory disturbances [36,51,52]. Of these agents, Dex is a very promising neuroprotective agent since it is already being used in clinical anesthesia practice. Dex may achieve its neuroprotection via its interactions with the alpha-2 adrenoreceptor [53,54], or by modulating the effect of other anesthetics on secondary messengers involved in the apoptotic cascade [55,56,57,58].

Recognizing Dex’s role in attenuating the effects of many anesthetics in neonatal animal models, Li *et al.* investigated whether Dex would have a similar neuroprotective effect during fetal exposure of propofol [36]. Dex attenuates the neurotoxic effects induced by *in utero* propofol exposure in three ways. Dex not only reduced propofol-induced caspase-3 activation and microglial cell response in the pups, but also improved the long-term neurocognitive function of the *in utero* propofol-exposed offspring [36]. These results provide additional evidence that fetal propofol exposure may cause neurotoxicity. These findings indicate that future studies using other reversal agents in a similar experimental animal model are warranted. Such *in vivo* animal studies can further extend the mechanisms causing neurotoxicity by fetal exposure to propofol. These studies may also help develop potential clinical applications to block or minimize any potential neurotoxic effects. 

## 6. Conclusions

Propofol, like most other general anesthetics, induce increased apoptotic activity in neonatal and fetal brains. These changes may lead to rewiring of the young brain, causing long-term behavioral changes and memory loss observed in animal studies. Thus, it is important to verify the mechanism(s) of propofol-induced neurotoxicity in young animals in order to modify the risk of neurodegeneration associated with propofol exposure. There are already some promising ideas that may serve to mitigate certain anesthetic-induced neurotoxicity including the addition of dexmedotomedine as a part of a general anesthetic regime.

## Figures and Tables

**Figure 1 brainsci-06-00011-f001:**
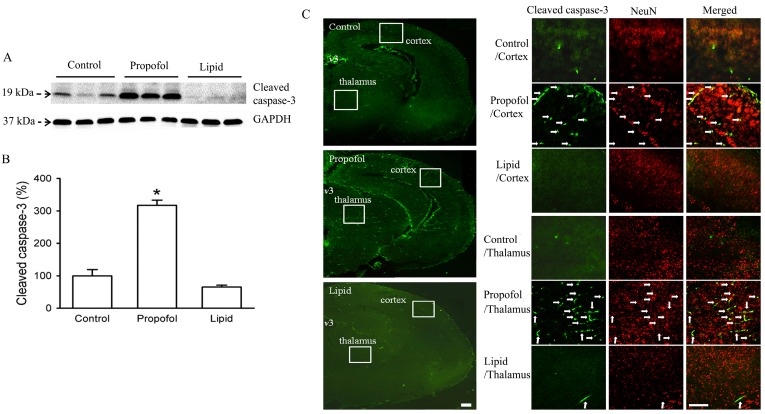
Propofol anesthesia for 1 h in pregnant rats induces caspase-3 activation in the brain tissues of fetal rats within 6 h. Propofol general anesthesia or intralipid (IV, 1 h) was administered to pregnant rats at the age of gestational days 20. After six hours, fetuses were removed, and fetal brain tissues were harvested and analyzed by Western blot*.* (**A**,**B**) Western blot data of cleaved caspase-3 in the fetal brain tissues exposed to propofol *in utero* and control condition (* *p* < 0.001, *n* = 4 fetuses/group); (**C**) Representative photomicrographs of the distribution of activated caspase-3-positive cells in the fetal brain tissues exposed to propofol *in utero* and control condition. There are sparse activated caspase-3-positive cells in the control fetal brain. However, activated caspase-3-positive neuronal profiles were abundant and heavily concentrated in regions such as the frontal cortex and thalamus in the fetal brains exposed to propofol. Double staining with antibody to cleaved caspase-3 (**green**) and antibody to NeuN, a neuron-specific nuclear protein (**red**), demonstrated that most of the cleaved caspase-3 positive cells were neurons (**white** arrows). *v*3, third ventricle. The small squares outlined in white in the right panel indicate the regions shown at higher magnification in the left panel. Scale bar = 100 µm in right panel; scale bar = 50 µm in left panel [36].

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
