# Peer review of "Propofol-Induced Neurotoxicity in the Fetal Animal Brain and Developments in Modifying These Effects—An Updated Review of Propofol Fetal Exposure in Laboratory Animal Studies"

_brainsci, 2016, doi:10.3390/brainsci6020011_

Round 1

Reviewer 1 Report

The authors try to review anaesthetic neurotoxicity with reference to propofol. However, several key works in this regards are missing and references are not update. The mechanisms of anaesthetic toxicity was not well supported by the arguments given in this review. Moreover a running text only in the review may not attract readers to go through the whole article. The authors should include some of their own illustrations, data tables, flow chart etc to illustrate their article to improve the quality of the work.

Author Response

Response: It is known that immature brain in animals may be very sensitive to general anesthetics, but there is controversy whether the period of brain sensitivity would include the fetal stage, especially the late fetal stage.  The review provides clear indication that excessive apoptotic activities are detected in fetal brain upon propofol exposure. Additionally, many research groups have focused on whether the apoptosis activity induced by propofol could be reversed or blocked with different agents. This would be of particular interest to clinicians since propofol is a widely used anesthetic and cannot be avoided in many cases.  As for illustration from our own research, all the data and table is published.

Additional comments( manuscript has been revised  after evaluated by Academic Editor):

Thank you for the one week extension for this revision.
We are very appreciative of the comments from both the academic advisor and the 2nd author reviewer.   The major focus of this review is whether propofol exposure during the fetal stage of the brain development may trigger apoptosis activity.   We have also explored the possibilities of minimizing the increased apoptotic activity induced with propofol.  The mechanism of propofol induced toxicity (apoptotic activity in the fetal brain) is currently under our active investigation.  We have updated several new references to support this idea in the  major revision.  We agree that additional illustration from our previous studies would be helpful to support this manuscript for reconsideration.
Figure 1:  Propofol exposure in the fetal brain leads to significant increases apoptotic activity.

Reviewer 2 Report

Brain Sciences

ISSN 2076-3425

“A review of propofol-induced neurotoxicity in the fetal animal brain and developments in modifying these effects-an updated review of propofol fetal exposure in laboratory animal”

Abstract

Line 4- I think that the beginning of this sentence must be improved: “Even less research has investigated…

-Suggestion: The title mentions “developments in modifying these effects….”; so, the last sentence of the abstract should be completed referring strategies to reverse the neurotoxic effects of propofol .

General comment

- During all the manuscript the authors mention many references used with the names of the respective authors, but they do not write the year of the publication.

Suggestions

A -Although the authors mention this reference in the manuscript “Fredriksson A et al., 2007”, I think it could be much more explored, especially concerning 2 issues: 1- propofol and GABA A receptors, and 2- the combined effect of propofol with NMDA antagonists because very often propofol is used in combination with another anesthetic.

B - The authors on page 4 (2nd parag, line 14/15) refer the neural inflammation on the developing brain. I think this issue could be developed using the reference number 30 (Kargaran et al. 2014).

C- Page 5- concerning the “reversal agents”, I suggest that the authors could use better the references mentioned in the text (46 and 47) to develop better the ideas of lithium and erythropoietin as chemicals used to protect anaesthetic-induced apoptosis.

Specific comment

Page 5- 2nd parag, line 4: I think it is not “attribute”, but “contribute” 

Author Response

 1. Line 4- I think that the beginning of this sentence must be improved: “Even less research has investigated…

-Suggestion: The title mentions “developments in modifying these effects….”; so, the last sentence of the abstract should be completed referring strategies to reverse the neurotoxic effects of propofol .

Response:  The sentence has been modified as suggested.

 2. During all the manuscript the authors mention many references used with the names of the respective authors, but they do not write the year of the publication.

 Response: I am sorry for the error. It has been fixed as suggested.

Suggestions

A -Although the authors mention this reference in the manuscript “Fredriksson A et al., 2007”, I think it could be much more explored, especially concerning 2 issues: 1- propofol and GABA A receptors, and 2- the combined effect of propofol with NMDA antagonists because very often propofol is used in combination with another anesthetic.

B - The authors on page 4 (2nd parag, line 14/15) refer the neural inflammation on the developing brain. I think this issue could be developed using the reference number 30 (Kargaran et al. 2014).

Response: The reference 30 has been removed.

C- Page 5- concerning the “reversal agents”, I suggest that the authors could use better the references mentioned in the text (46 and 47) to develop better the ideas of lithium and erythropoietin as chemicals used to protect anaesthetic-induced apoptosis.

Response: Two more reference have been added.

Specific comment

Page 5- 2nd parag, line 4: I think it is not “attribute”, but “contribute”

Response: It has been modified as suggested.

Round 2

Reviewer 1 Report

The revised report is better that the previosu ones. however, some key references are still missing where propofol induced neurotoxicity or abnormal cellular function is reproted earlier (see below):

Propofol promotes blood-brain barrier breakdown and heat shock protein (HSP 72 kd) activation in the developing mouse brain.

Sharma HS, Pontén E, Gordh T, Eriksson P, Fredriksson A, Sharma A.

CNS Neurol Disord Drug Targets. 2014;13(9):1595-603.

PMID:

25106637

Select item 215400612.

Neonatal exposure to propofol affects BDNF but not CaMKII, GAP-43, synaptophysin and tau in the neonatal brain and causes an altered behavioural response to diazepam in the adult mouse brain.

Pontén E, Fredriksson A, Gordh T, Eriksson P, Viberg H.

Behav Brain Res. 2011 Sep 30;223(1):75-80. doi: 10.1016/j.bbr.2011.04.019. Epub 2011 Apr 21.

PMID:

21540061

Select item 177212453.

Neonatal exposure to a combination of N-methyl-D-aspartate and gamma-aminobutyric acid type A receptor anesthetic agents potentiates apoptotic neurodegeneration and persistent behavioral deficits.

Fredriksson A, Pontén E, Gordh T, Eriksson P.

Anesthesiology. 2007 Sep;107(3):427-36.

PMID:

17721245

I wonder how these reprots could not come acroos teh author's literature search? Anyway the discussion may need to be revised based on some key facts discussed in these papers.

Author Response

Response to the reviewer (3/4/2016):

Dear Jason and second reviewer: 

We are very appreciate the commend and suggestions.  This review paper is focused on fetal propofol exposure and possible blocking methods in the studies of propofol induced neurotoxicity.  We citied the research work by Fredriksson et al (2007) in our neonatal propofol exposure review (ref 19).  This lab work compared low dose (10mg/kg) vs. high dose (60mg/kg) of propopol, GABAa vs NMDA agents, or combination of propofol and ketamine in the young developing brain. 

We also reviewed the review’s suggestion, and agree that “propofol affect the BDNF expression and promote the blood brain barrier break down” provide further evidence that propofol exposure in neonate brain may contribute to neurotoxicity studies in neonate animal. We add those two lab works to the ref, 55 and 56.

We add sentence in section of profopol induced neurotoxicity in neonate animal brain.  We believe those studies not only shed light on possible mechanism of propofol induced neurotoxicity, but also provide a great evidence for long-term behavioral change in animal studies.

Sincerely

Ming Xiong, MD., Ph.D.

NJMS-Rutgers
